# Intersectional Fairness Score : the overlooked but far-reaching choice of aggregation design

## Abstract

Fairness assessment in AI is essential for building responsible models. Traditionally, it focuses on two demographic groups situations, but real-world complexity requires considering intersectionality [2]. This work explores how to aggregate bias measurements across multiple subgroups into one single score—a critical step often overlooked. We first show that the choices made in aggregation design (norm, maximum, probabilistic approaches, etc.) can significantly influence results, leading to divergent or even conflicting conclusions. We identify and analyze the various possible methods, highlighting their ethical implications and providing a first framework of criteria to guide their selection based on context. Our goal is to foster interdisciplinary discussion on this often-neglected step, aiming for a fairer, more informed and transparent evaluation of intersectional biases in AI.

## 1 Introduction

Fairness assessment in artificial intelligence (AI) models is a critical step toward developing responsible, trustworthy, and ethically sound systems. As AI increasingly influences high-stakes decisions across diverse domain, the need for rigorous evaluation of biases and fairness becomes paramount. Existing literature has established a broad framework for fairness metrics [7], highlighting the complexity of defining and operationalizing fairness. These efforts often focus on binary scenarios, where fairness notions are straightforwardly applied to two groups distinguished by a unique attribute. However, real-world applications frequently involve multiple sensitive attributes requiring an intersectional approach of bias to consider the combinatorial way discrimination occurs.

Intersectionality, originally conceptualized within social sciences, emphasizes the interconnected nature of social categorizations such as race, gender, and class, which produce overlapping systems of discrimination or privilege. In the context of AI fairness, addressing intersectionality involves moving from a binary two-dimensional problem to a multi-dimensional one. This transition comes with new particularities and additional steps in the process of bias assessment. We claim that these steps have been overlooked due to insufficiently meticulous reuse of existing tools developed for the binary case.

This work aims to shed light on the specific step of subgroups measure aggregation strategy to produce a comprehensive fairness measure. We argue that this step implies a series of technical choices that are embedded with ethical considerations. After an identification and an analysis of various fairness score design possibilities (Section 3), we demonstrate how different choices can lead to markedly divergent fairness assessments (Section 4). Practical and intuitively understandable examples illustrate the profound impact of these decisions, emphasizing the importance of informed, context-sensitive design rather than default or arbitrary selections. Section 5 opens a discussions on some guidelines that could help choose the appropriate technical tool depending on the results of the ethical analysis of the model building's context.

Submitted to 39th Conference on Neural Information Processing Systems (NeurIPS 2025). Do not distribute.

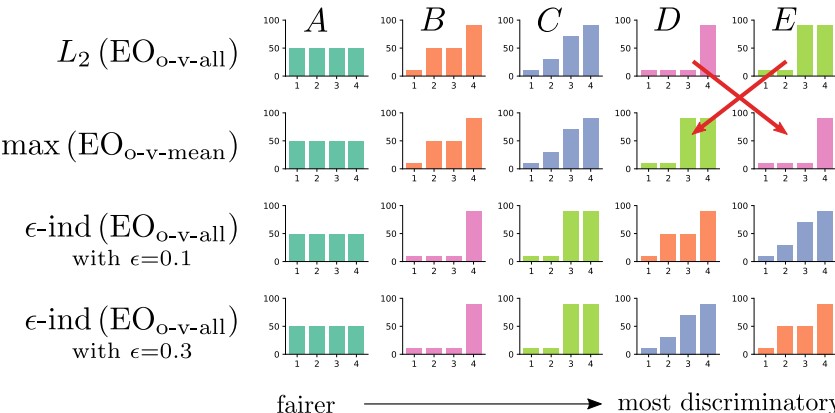

Figure 1: Rankings obtained by different aggregation methods on the same set of values for subgroups. By choosing a different aggregation method, we can reverse what is considered fair or dicriminatory.

Our contribution underscores the necessity of making explicit the design choices involved in intersectional fairness assessment. By raising awareness of this critical stage, we aim to foster more transparent, equitable, and contextually appropriate AI systems, ultimately advancing the field toward more responsible AI development.

## 2 Related Work

**Fairness as part of AI ethics** Fairness in AI require choices in defining the assessment framework [9]. These are not only technical, but ethical in essence. As ethical choices, they address questions that do not have a universal correct and obvious answer, but rather require careful consideration in order to ultimately determine the best possible solution. These elements encompass the context in which the AI model will be used, among which cultural habits, values, political ideologies and societal choices. These should be collaborative discussed, in working groups comprising individuals from a range of social backgrounds and different academic disciplines (law, sociology, philosophy, formal sciences, etc.), informed by comprehensive technical assessments of potential fairness issues [10].

**The various levels of technical choices in bias assessment that involve ethical considerations** Aggregation design is not the first technical and ethical choice. First, human characteristics on which discrimination must not be based have to be identified [4]. In AI fairness, the term *sensitive attribute* is used to refer to these characteristics [8]. Whilst this step may appear to be an immediate result, it is rare for a justification to be provided for the reasons behind their selection.

Secondly, If we agree that discrimination is prejudicial differential treatment, how does this translate in practical terms in the case of an AI model prediction? [1] If we focus on the case of binary classification models, is it to obtain the disadvantageous prediction? is it to obtain this prediction knowing that we should have obtained the opposite one ? The extensive range of possibilities has resulted in the generation of numerous *notions of fairness* in the state of the art [7]. Among the best known and most frequently used : Statistical Parity (SP), Equalized Odds (EOd) and Equal Opportunity (EO).

## 3 Problem Setting

Let $D$ be a dataset of instances $(\mathbf{X}, \mathbf{A}, Y) = (x_1, \ldots, x_m, a_1, \ldots, a_{n-m}, y)$, each corresponding to an individual. Among the features, $\mathcal{A} = \prod_{i=m+1}^{n} \mathcal{A}_i$ denotes the *sensitive attributes*, i.e the features for which discrimination should not occur. We denote the number of sensitive attributes considered as $|\mathcal{A}|$. When $|\mathcal{A}| > 2$, intersectionality occurs. Based on the sensitive attributes, we define *demographic groups* or *subgroups* in intersectional setups, denoted as $g \in G$, where each group is characterized by individuals sharing the same values for the sensitive attributes. The total

number of groups considered is denoted by $|G|$. We distinguish two main cases: $|G| = 2$, the binary scenario on which existing literature mostly focuses, and $|G| > 2$, the non-binary scenario.

For a classifier $M$ learned using a machine learning algorithm, group fairness is usually measured by choosing a metric (e.g. the probability of being correctly classified, or the probability of being assigned the advantageous outcome), corresponding to the previously chosen *fairness notion*, that quantifies how much its prediction differs depending on whether an instance belongs to a protected group or not. Even though our work is generic and can be extended to different choices of fairness notion/metric, for ease of exposure, we focus on the Equalized Odds metric $EO(M) = P(M(x) = 1|Y = 1, A = 1) - P(M(x) = 1|Y = 1, A = 0)$ that quantifies the difference in the odds of obtaining the positive outcome $M(x) = 1$ provided that the true class $Y$ is 1, depending on the protected attribute $A$.

**Binary case**  Provided that the outcome $M(x) = 1$ is judged favorably, the metric directly compares the (potential) benefit of belonging to subgroup $A = 1$: $P(M(x) = 1|Y = 1, A = 1)$ over subgroup $A = 0$: $P(M(x) = 1|Y = 1, A = 0)$ by a simple subtraction since there are only 2 quantities to compare, which immediately results in a single fairness score. When comparing any 2 models $M_1$ and $M_2$, we can tell which model is fairer than the other by comparing the score (a scalar).

**Intersectionality: the need for summarizing many measurements.**  Intersectionality implies a setup where the number of subgroups $|G| > 2$. We obtain a benefit score for each subgroup. Comparison of benefit for every pair of subgroups results in $\frac{|G|(|G|-1)}{2}$ measurements, that we would like to *summarize in a global fairness score* by aggregating measurements into a single scalar. The choice of aggregation method is essential, as the subtle differences between aggregation methods can imply different conclusions. Model $M_1$ can be deemed more fair than model $M_2$ by a choice of aggregation method, or less fair by a different choice.

While there are certain applications (e.g., healthcare) where improving the utility of the worst-case groups is an important goal, other applications can be required by law to ensure more parity for all subgroups. In the following, we explicitly list the possible design choices when applying fairness assessment to an intersectional setup, and describe different mathematical tools to summarize $|G|$ measurements into a single scalar value.

### 3.1  First design choice: comparison type

In the **one-vs-all** approach, we form the vector of differences between all $\frac{|G|(|G|-1)}{2}$ possible pairs $(i, j)$ of subgroups.

$$EO_{\text{o-v-all}}(M, i, j) := P(M(x) = 1|Y = 1, A = A_i) - P(M(x) = 1|Y = 1, A = A_j)$$

In the **one-vs-mean** approach (e.g. in [3]), the value measured on one subgroup is instead compared to the others through their mean value and we use the $|G|$ vector of differences to the averaged benefit.

$$EO_{\text{o-v-mean}}(M, i) := P(M(x) = 1|Y = 1, A = A_i) - P(M(x) = 1|Y = 1)$$

In these first two approaches, subtraction is used but we could instead use ratios.

The **all-in-one** approach is slightly different because it is not founded on an individual analysis but directly encompasses all measures immediately leading to the final result of a single score. It studies the dependence of the variable of interest, i.e. the variable considered by the chosen fairness metric, on the random vector of sensitive attributes, as measured by an independence criterion (IC) such as the Mutual Information [5, 6] or the Hilbert Schmidt Independence Criterion.

$$EO_{\text{all-in-one}}(M) := IC(M(x) = 1, A \mid Y = 1)$$

### 3.2  Second design choice: aggregation method

In the one-vs-all and one-vs-mean approaches, it is then necessary to aggregate the multiple values obtained, which can be achieved using either:

$L_q$ **norms**: Let $q \in \mathbb{R}^+$, a $q$-**norm** is defined for any vector $(x_1, x_2, ..., x_n)$ by

$$L_q = (\sum_{i=1}^{n} \mid x_i \mid^q)^{1/q}$$

$L_q$-norms provide a flexible measure of the magnitude of a vector, where larger values of $q$ penalize the largest components, until the limit $q \to \infty$ where $L_\infty := \max_i(x_i)$:

- **For q = 1**: a uniform weight is given to all subgroups [3];
- **For q = 2**: the norm highlights significantly discriminated subgroups more strongly, as it emphasizes higher values;
- **For q → ∞**: the norm is dominated by the most discriminated subgroup, indicating the **worst case** scenario.

**Ordered Weighted Averaging**: OWA consists of calculating the weighted average of a vector by weighting its elements according to their rank. Let $(x'_1, x'_2, ..., x'_n)$ be values of $(x_1, x_2, ..., x_n)$ ranked by increasing order, and $(w_1, w_2, ..., w_n)$ be a set of weights.

$$OWA := \sum_{i=1}^{n} w_i x'_i$$

This method also allows to adjust the sensitivity of the aggregate measurement to extreme values or overall distribution, with even greater flexibility than that allowed by $L_q$ norms. $(w_1, w_2, ..., w_n)$ are hyperparameters that broaden the possibilities and make the method fully customizable. It notably encompasses measure of **minimum**, **maximum** and **mean**.

**Threshold indicators**: Let $\varepsilon \in [0, 1]$. For any vector $(x_1, x_2, ..., x_n)$, the *$\varepsilon$-threshold indicator* counts the number of values that are greater than $\varepsilon$:

$$\varepsilon - ind := \frac{1}{n} \sum_{i=0}^{n} \mathbb{1}(x_i > \varepsilon)$$

The hyperparameter $\varepsilon$ sets the threshold height. For small $\varepsilon$ values, the priority is to avoid gaps, no matter how large or small. For large $\varepsilon$ values, the focus is more on avoiding very large gaps. This focuses attention on whether a certain threshold has been exceeded in the measured gaps. Unlike previous methods, it does not consider the magnitude of the gaps. The salient consideration is the frequency with which biases exceed a substantial magnitude to be considered. This places greater emphasis on minimizing the occurrence of bias rather than reducing it to the lowest possible level.

### 3.3 Variations

The possibilities listed in the previous sections lead to methods which can be adapted to specific needs by slightly modifying them in different ways. One option is to weight measurements made by subgroups according to the size of their sample. This is useful when samples are too small and may not be representative. Weighting will enable the exclusion of outliers that could have a significant impact even though they are in fact incorrectly estimated.

On the contrary, there is in fairness a propensity to focus on small samples, which often correspond to subgroups less visible and therefore discriminated against. It may be desirable to give more weight to these groups. In this case, weighting by the inverse of the sample size might be considered.

## 4 Illustrative cases

**Different aggregation methods implicitly imply different rankings**   As a first illustrative case, for 4 choices of aggregation methods, we rank the same 5 distributions of benefits over subgroup as measured by their aggregated score (Figure 1). All 4 methods deem distribution $A$ fairer among all since all subgroups receive the exact same benefit, but even though these 5 distributions all span the same range of benefit, and have the same mean benefit $= 50$ (except for distribution $D$), changing aggregation alters the overall ranking. For instance (red arrows), distribution $D$ and $E$ are exchanged by switching from aggregation method $L_2$ ($\text{EO}_{\text{o-v-all}}$) to $\max$ ($\text{EO}_{\text{o-v-mean}}$).

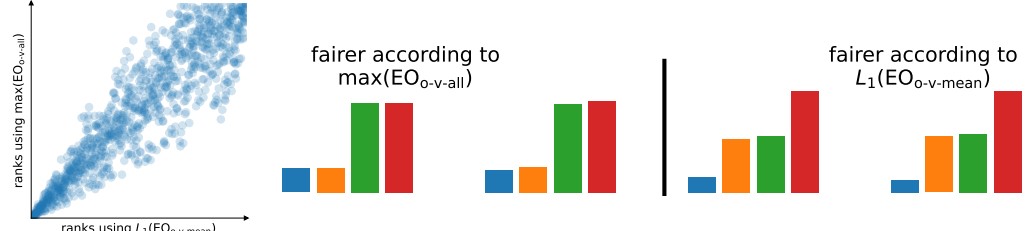

Figure 2: Pairwise comparison of aggregation methods $L_1\left(\text{EO}_{\text{o-v-mean}}\right)$ and $\max\left(\text{EO}_{\text{o-v-all}}\right)$. **(left)** rank correlation and **(right)** distributions that produce most different rankings.

**Systematically quantifying pairwise difference between aggregation methods**   We randomly sample $1000$ distributions of benefits for 4 subgroups with equal mean $= 50$. We measure their fairness score using aggregation methods $L_1\left(\text{EO}_{\text{o-v-mean}}\right)$ and $\max\left(\text{EO}_{\text{o-v-all}}\right)$, and compare the rankings. In Figure 2 (left), we observe that overall, these produce quite correlated rankings (Kendall's $\tau$ rank correlation $= 0.75$). We then identify the distributions for which the rankings change the most (Figure 2 right). Highlighting these discrepancies reveals the implications of the chosen aggregation method. As we argue, this choice should be emphasized to encourage an informed discussion.

## 5   Discussions and Conclusions

In Sections 3 and 4, we highlighted the impact of the fairness score design on the valued patterns of biases distributions. This intersectional fairness score can ultimately serve two purposes: first, as a simple measure of bias in a dataset or model (fairness assessment); and second, as an indicator to guide bias mitigation, for example by regularizing the learning objective function. In the second case, this means that the bias mitigation protocol will cause the distribution of biases in the model to tend towards those that obtain the best rankings by applying the chosen aggregation design. Consequently, the score must be designed in line with how stakeholders define fairness.

### 5.1   Some guidelines

To facilitate the understanding and implementation of these fairness tools in AI models training, it is essential to link analytical studies and observations with practical model user expectations. To this end, we present preliminaries sets of guidelines that may assist in this decision. These initial elements will require further study in future work.

**Trade-off between focusing on amplitude of gaps or frequencies of gaps.**   In the first case, $q$-norms and OWA are recommended while $\varepsilon-$threshold are better suited for second option.

**Worst case or global amount of bias.**   This criterion is achieved through correct adjustment of hyperparameters. High $q-$ and $\varepsilon-$values will both lead to focus on worst cases, and vice versa.

**Distribution of biases leading to isolated cases, or formation of blocks of groups treated in the same way.**   The choice of either *one-vs-mean* or *one-vs-all* approach enables the transition towards the first situation or the second one, respectively.

### 5.2   Multidisciplinary discussion is key

Addressing the ethical challenges posed by advanced technological tools such as AI models requires more than technical solutions alone. The role of the researcher in fairness in machine learning cannot go beyond providing these technical tools for fairness assessment in intersectional setups, and analyzing their implications to be discussed transparently. Subsequently, a multidisciplinary dialog is required to bring together expertise from formal sciences and social sciences, as well as collective societal discussion in the general public in cases that require political choices. This would lead to the best informed option to answer the ethical goal of assessing and mitigating biases, and integrate the corresponding tools in our mathematical models and AI systems.

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
