# OpenReview forum: "Intersectional Fairness Score : the overlooked but far-reaching choice of aggregation design"
_EurIPS.cc/2025/Workshop/UPLB — UPLB2025_

### Official Review · Reviewer_ivh1 · 2025-10-26

**Rating:** 4
**Confidence:** 3

**Review:**

The paper deals with the problem of intersectional fairness assessment in AI. This is the evaluation of models trained on distributions with, among others, multiple sensitive attributes, i.e. features that should not impact the classification outcome, for ethical reasons. When more than 2 subgroups, based on the values of their sensitive attributes, are present, fairness assessment involves a combinatorial number of comparisons, which leads to the need of aggregation methods. The paper compares 3 families of them ($L_q$ norms, ordered weighted averaging and threshold indicators), showing that their choice can lead to different fairness rankings for the same set of unequal distributions of benefits. It gives some guidelines on why choosing a method rather than another. It advocates for more transparent choices of protocol, making explicit the ethical implications behind them.

**Strengths**

The problem of intersectional fairness assessment in AI is timely an in-scope with the UPLB workshop. The paper is well written, the setting clearly stated and the results explained.

**Weaknesses**

- The technical contribution of the paper seems to be the numerical tests in Figure 1 and 2, which report fairness scores of synthetic distributions of benefits, for direct comparison. The theoretical framework is limited to the standard definitions of multigroup comparison and aggregation methods in Problem setting. I do not find the results and the guidelines in conclusion particularly interesting or useful. For example, that high-$q$ $L_q$ norm focuses on worst cases is clear from the definition. That different aggregation methods lead to different rankings of random distributions, despite the scores being quite correlated, is also hardly surprising. In this respect, the paper reads more as a position paper than a workshop contribution.
- Despite advocating for an interdisciplinary approach, the “Related work” section is rather poor. I understand that the limited space granted by the format is a factor, but at least the closest and more recent works addressing the problem of testing fairness in ML should  have been referenced. See non-exhaustive list of examples below. Moreover, rather than addressing directly the problem of fairness assessment in AI, the paper is more about the much broader problem of measuring inequality in distribution/aggregation measures, but no reference has been made on that.
- Figure 2 compares two different aggregation methods ($L_1$ vs max) evaluated on different metrics (one-vs-mean vs one-vs-all). As the scope of the paper is to investigate different aggregation designs, it could possibly be better to compare aggregation methods of the same measure, to disentangle the effect of the choice of metrics from that of the aggregation.

Castelnovo et al. “A clarification of the nuances in the fairness metrics landscape.” Sci Rep 12, 4209 (2022)

Gohar, Cheng. “A Survey on Intersectional Fairness in Machine Learning: Notions, Mitigation, and Challenges.” (2023)

Purificato et al. “Toward a Responsible Fairness Analysis: From Binary to Multiclass and Multigroup Assessment in Graph Neural Network‑Based User Modeling Tasks” (2023)

---

### Decision · Program_Chairs · 2025-11-03

Accept (Poster)